# Improved Breakdown Strength and Restrained Leakage Current of Sandwich Structure Ferroelectric Polymers Utilizing Ultra-Thin Al_2_O_3_ Nanosheets

**DOI:** 10.3390/nano13212836

**Published:** 2023-10-26

**Authors:** Yi Zeng, Hao Pan, Zhonghui Shen, Yang Shen, Zhifu Liu

**Affiliations:** 1Faculty of Printing Packaging Engineering and Digital Media Technology, Xi’an University of Technology, Xi’an 710048, China; 2State Key Laboratory of New Ceramics and Fine Processing, School of Materials Science and Engineering, Tsinghua University, Beijing 100084, China; 3State Key Laboratory of Advanced Technology for Materials Synthesis and Processing, Center of Smart Materials and Devices, Wuhan University of Technology, Wuhan 430070, China; 4CAS Key Laboratory of Inorganic Functional Materials and Devices, Shanghai Institute of Ceramics, Chinese Academy of Sciences, Shanghai 201899, China

**Keywords:** nanosheet, Al_2_O_3_, P(VDF-HFP), breakdown strength, leakage

## Abstract

Flexible capacity applications demand a large energy storage density and high breakdown electric field strength of flexible films. Here, P(VDF-HFP) with ultra-thin Al_2_O_3_ nanosheet composite films were designed and fabricated through an electrospinning process followed by hot-pressing into a sandwich structure. The results show that the insulating ultra-thin Al_2_O_3_ nanosheets and the sandwich structure can enhance the composites’ breakdown strength (by 24.8%) and energy density (by 30.6%) compared to the P(VDF-HFP) polymer matrix. An energy storage density of 23.5 J/cm^3^ at the ultrahigh breakdown strength of 740 kV/mm can be therefore realized. The insulating test and phase-field simulation results reveal that ultra-thin nanosheets insulating buffer layers can reduce the leakage current in composites; thus, it affects the electric field spatial distribution to enhance breakdown strength. Our research provides a feasible method to increase the breakdown strength of ferroelectric polymers, which is comparable to those of non-ferroelectric polymers.

## 1. Introduction

Low-cost electronic and power systems tend to use dielectric capacitive polymers with lightweight, low-cost, and high-energy storage density [1,2,3]. Generally, a dielectric material’s energy density (*U*_e_) is determined by the electric displacement (*D*) upon an applied electric field (*E*), which can be illustrated as $U_{e}=\int_{D_{max}}^{0}EdD$ [4]. Therefore, improved *D* and *E* can increase energy density in dielectric capacitive polymers. Moreover, a moderate permittivity is needed to refrain from the early electric displacement saturation at a field much lower than the breakdown electric field [1]. Generally, dielectric materials of capacitors can be categorized as inorganics (bulk or film ceramics) and organics (polymer). Inorganics can deliver a relatively higher energy efficiency and energy density [5,6,7,8,9]. However, their low breakdown strength and non-flexible properties limited the application field. Organics, on the contrary, represent lower cost and easier fabrication on a large scale, which makes them more commonly used in commercial applications.

The commercially available polymer film capacitors, i.e., biaxially oriented polypropylene (BOPP) as a non-polar polymer, usually possess high breakdown strengths over 700 MV/m but low dielectric constants (under 5 for BOPP); thus, it has low energy densities of 1∼2 J/cm^3^ [10,11]. Contrary to non-polar polymers, polar polymers such as ferroelectric-based polymers (e.g., poly (vinylidene difluoride), PVDF) usually have high energy density (>10). However, ferroelectric polymers’ C-F bonds have strong orientation polarization, which causes low breakdown strength in polymers [12]. To improve the breakdown strength, energy storage efficiency, dielectric permittivity, and capacitive energy density, polar polymers are extensively investigated with organic blends [13,14], multilayer design [15,16] or inorganic fillers of different dimensions [2,17,18,19], such as nanoparticles and quantum dots (0D) [20,21,22,23,24,25], nanofibers, nanorod array (one-dimensional, 1D) [26,27,28,29], as well as nanoplates and nanosheets (two-dimensional, 2D) [30,31,32,33,34].

Polymer matrices with nanofillers usually have an excellent overall performance; therefore, this strategy has been used to improve the composites’ energy storage properties. However, the electric field distribution is inhomogeneous in some regions, such as when electric field aggregation in the electric field direction at nanoparticles’ shoulders occurs. When the breakdown region comes into contact with nanoparticles, it has to detour around the nanoparticles to go forward. Consequently, the electric field breakdown strength of the polymer matrix with nanoparticles is low [35]. For the above reasons, in our previous study, the ferroelectric polymer matrix obtained excellent energy storage performance after switching to nanosheets as an alternative to nanoparticles [28].

Two-dimensional nanosheets such as Boron Nitride (BN), clay, TiO_2_, NaNbO_3_, and ZrO_2_ et al. [30,31,32,33,34] are usually used in nanocomposites to improve the dielectric properties and storage density. Alumina can also be used to regulate the dielectric properties of materials [36]. However, it is rather difficult to obtain few-layer or single-layer 2D nanosheets, and the complicated processing procedure (such as liquid-phase exfoliation) is limited in practical applications.

Here, we demonstrate an ultra-thin 2D insulating material, Al_2_O_3_ nanosheets, which can be achieved through a simple method with a hydrothermal process instead of a complicated peeling strategy. It is revealed that the incorporation of the 2D Al_2_O_3_ nanosheets, combined with structure modulation, can remarkably affect ferroelectric-based polymers’ electric field spatial distribution. Therefore, remarkable improvements in the breakdown strength and energy density of composites are achieved.

## 2. Experiment

### 2.1. Preparation of Samples

Figure 1 shows the preparation process diagram of ultra-thin Al_2_O_3_ nanosheets (2D-Al_2_O_3_) and P(VDF-HFP)/2D-Al_2_O_3_ nanocomposites. Firstly, AlCl_3_·6H_2_O, NaOH, NH_3_·OH (Sinopharm Chemical Reagent Co., Ltd., Shanghai, China) were separately placed in deionized water in a concentration of 1 M solution(A); then the NaOH and NH_3_·OH mixed solution was dropped slowly into A solution until the pH became over 8, and then it was transferred to a reactor with Teflon lining. The reactor was placed in an oven at 200 °C for over 10 h; after the reaction, the reactor was cooled naturally, and the precipitate was washed with deionized water and ethanol repeatedly. Then, the precipitate was dried, obtaining the ultra-thin pre-nanosheet powders, which were calcined at 700 °C for 1 h, obtaining ultra-thin Al_2_O_3_ nanosheets. The structure of the calcined nanosheets is very stable because it forms a gamma phase of Al_2_O_3_ (Figure 2 XRD images of 2D-Al_2_O_3_ nanosheets).

Secondly, Al_2_O_3_ nanosheet powders were dispersed in N,N-dimethylformamide (DMF) and acetone (Sinopharm Chemical Reagent Co., Ltd., Shanghai, China). Stirring the mixture over 12 h makes the mixture homogeneous and stable. The Al_2_O_3_ nanosheets and P(VDF-HFP) (with 10 wt% HFP, Kynar Flex 2801, Arkema, Colombes, France) were dispersed into N,N-dimethylformamide (DMF); the mixture was stirred over 15 h for homogeneousness, and then the precursor sol was produced, which was used for the next electrospinning process. The viscosity of the sol was regulated by P(VDF-HFP), and then the sol was placed in the injector; the electrospinning process was under 1.3 kV cm^−1^ electric field. After electrospinning, these electrospun fibers were layered by sandwich structure to the next hot-pressing process. The temperature of the hot-pressing process is 200 °C, 30 min. The last process is reheating the composites to 240 °C (for 7 min) and quenching them in 0 °C water. All the preparation processes are repeated more than three times to ensure the repeatability of the process.

A series of sandwich structures in P(VDF-HFP)/2D-Al_2_O_3_ nanocomposites are prepared by the electrospinning process; the trilayered nanocomposites are named “×0×”. “×” is the volume fraction of 2D-Al_2_O_3_ nanosheets layer, and “0” is the pure P(VDF-HFP) layer. For example, “1-0-1” refers to trilayered films: 1 vol.% 2D-Al_2_O_3_ nanosheet layers are in the outer layer, and a pure P(VDF-HFP) layer is in the middle layer.

### 2.2. Characterization

The microstructure of Al_2_O_3_ nanosheets was characterized by an X-ray diffractometer (XRD, D8 Advance, Bruker, Mannheim, Germany) and high-resolution transmission electron microscopy (HRTEM, JEOL2011, JEOL, Akishima, Japan). The nanocomposites were characterized by scanning electron microscopy (SEM, ZEISS MERLIN compact, ZEISS, Jena, Germany).

For the measurements of dielectric properties, copper electrodes (2.5 mm in diameter and 50 nm in thickness) were deposited on top of the nanocomposites as the top electrodes, and aluminum foil was placed at the bottom electrodes. This research used an HP 4294A precision impedance analyzer (Agilent Technologies, Inc., Santa Clara, CA, USA) to measure the nanocomposites’ dielectric properties at room temperature; the measurement frequency ranged from 10^2^ to 10^7^ Hz. A Premier II ferroelectric test system (Radiant Technologies, Inc., Albuquerque, NM, USA) was used to measure samples’ electric displacements-electric field (D-E) loops at 10 Hz. Samples’ electric field breakdown strength was measured by the Dielectric Withstand Voltage Test (Beijing Electro-Mechanical Research Institute Supesvoltage Technique, Beijing, China); the current limitation parameter was 5 mA, and the ramping rate was 200 V/s. Each composite film includes over 30 tested points to achieve repeatability of properties. The defects introduced during sputtering copper electrodes or the preparation of composites will cause low properties of composites; however, this probability is approximately less than 5 percent.

### 2.3. Model of Phase-Field

This research uses a phase-field model; its phase-field variable *η* (*r*, *t*) is continuous. This phase-field variable depends on time and space [35]; therefore, it can be described as the P(VDF-HFP)/2D-Al_2_O_3_ nanocomposites’ electrostatic damage process. When the phase-field variable is “1”, it indicates the breakdown region; when it is “0”, it indicates the non-breakdown region. Between these two regions, there is a transition region, which is the interface region. Because sandwich structure P(VDF-HFP)/2D-Al_2_O_3_ nanocomposites is a dielectric inhomogeneous system, its total free energy should include the electric field, the interface, and the phase separation synergistic contributions. It can be written as
(1)$$F=\int_{V}\left[f_{sep}\left(\eta \left(r\right)\right)+\frac{1}{2}\gamma {\left|\nabla \eta \left(r\right)\right|}^{2}+f_{elec}\left(r\right)\right]dV$$

Sandwich structure P(VDF-HFP)/2D-Al_2_O_3_ nanocomposites’ breakdown phase evolution process is simulated by a modified Allen–Cahn equation:(2)$$\frac{\partial \eta \left(r,t\right)}{\partial t}=-L_{0}H\left(f_{elec}-f_{critical}\right)\left[\frac{\partial f_{sep}\left(\eta \right)}{\partial \eta \left(r,t\right)}-\gamma \nabla^{2}\eta \left(r,t\right)+\frac{\partial f_{elec}\left(r\right)}{\partial \eta \left(r,t\right)}\right]$$
where *L*_0_ is the kinetic coefficient and relates to the P(VDF-HFP)/2D-Al_2_O_3_ nanocomposites’ interface mobility. *H*(*f*_elec_ − *f*_critical_) is the Heaviside unit step function (*H*(*f*_elec_ < *f*_critical_) = 0 and *H*(*f*_elec_ > *f*_critical_) = 1). *f*_critical_ is a material constant that depends on position; it relates to the maximal energy density of each component in the P(VDF-HFP)/2D-Al_2_O_3_ nanocomposites.

## 3. Results and Discussion

Here, our work introduces other nanosheets (2D-Al_2_O_3_ nanosheets) with a diameter of less than 100 nm than has ever been reported and are used to replace nanoparticles as the insulating fillers in nanocomposites, studying the effect of 2D-Al_2_O_3_ nanosheets on the electric field distribution in sandwich structure P(VDF-HFP)/2D-Al_2_O_3_ nanocomposites. Figure 2a,c show the microstructure and X-ray diffraction (XRD) of 2D-Al_2_O_3_ nanosheets and high-resolution transmission electron microscopy (HRTEM). From Figure 2a, the nanosheets are well crystallized with a uniform morphology, and their crystal size mainly distributes under 100 nm.

The image of the trilayered P(VDF-HFP)/2D-Al_2_O_3_ nanocomposites’ microstructure cross-section is shown in Figure 2b. H is the thickness of the trilayered film, which is about 12.5 μm, and h is the thickness of the P(VDF-HFP)/2D-Al_2_O_3_ layer, which is about 1.25 μm. It shows the homogeneity of P(VDF-HFP)/2D-Al_2_O_3_ nanocomposites in different layers. Meanwhile, the interface between layers is without structure defects, for instance, voids and pores. The above results mean that the optimized sandwich structure preparation process has greatly improved the quality of P(VDF-HFP)/2D-Al_2_O_3_ nanocomposites, thereby providing a good foundation for improving the properties of dielectric and energy storage.

Ferroelectric polymer matrix P(VDF-HFP) has a relatively high electric field breakdown strength, while it has a good open circuit breakdown property. Hence, it has been potentially used as a dielectric for power capacity. Aluminum oxide is a fine, favorable insulating material; therefore, it has been commonly used in capacity polymers as a kind of dielectric filler. To sum up, in this work, sandwich structure P(VDF-HFP)/2D-Al_2_O_3_ nanocomposites’ energy storage properties should include the electric field, the interface, and the phase separation between the P(VDF-HFP) and aluminum oxide synergistic contributions.

Usually, the energy storage performances are multiply determined by insulating properties, such as capacitances (Figure 3a), dielectric loss (tanδ) (Figure 3b), and breakdown strength (Figure 4). As shown in Figure 3, the introduction of 2D-Al_2_O_3_ nanosheets into the ferroelectric polymer matrix can maintain relatively high insulating performance but cannot restrain the dissipation occurring at high frequency, which is mainly caused by the resonance of the ferroelectric polymer matrix.

Figure 4 shows the comparison of ferroelectric and energy storage performances between sandwich structure P(VDF-HFP)/2D-Al_2_O_3_ nanocomposites and the P(VDF-HFP) matrix under their breakdown electric fields. However, due to the intrinsic electric polarization of P(VDF-HFP), it only bears restricted energy density. In this work, the sandwich structure of ×0× nanocomposites significantly enhanced the electric field strength and electric displacement of the ferroelectric polymer matrix. It is calculated that the energy density of ×0× nanocomposites is always higher than that of the P(VDF-HFP) polymer under the breakdown electric field. The energy density of P(VDF-HFP) is about 18 J/cm^3^ at 600 kV/mm, while the energy density of 3-0-3 nanocomposites is about 23.5 J/cm^3^ at over 700 kV/mm. The higher energy density of 3-0-3 nanocomposites is attributed to its higher breakdown strength (*E*_b_).

For a more detailed comparison between ×0× trilayered nanocomposites, P(VDF-HFP), and other references reporting ferroelectric composites’ energy storage properties, Figure 5 itemizes their discrepancy in the electric field breakdown strength and discharged energy density. Obviously, all ×0× nanocomposites have outstanding properties in the electric field breakdown strength and discharged energy density.

The 3-0-3 nanocomposite’s electric field breakdown strength increased by 24.8% over that of pure P(VDF-HFP) film. The discharged energy density of it is 30.6% higher than that of the pure P(VDF-HFP) film. Both increased breakdown strength and relatively lower energy loss of the ×0× nanocomposites cause an increase in discharged storage density.

To further study which factor affects the efficiency of ×0× nanocomposites, we compare the residual polarization values of ×0× nanocomposites with field strength in Figure 6. All the nanocomposites have a low remnant displacement of <0.5 μC cm^−2^ until the electric fields increase to 200 kV/mm because it did not have time to obtain a phase transformation in P(VDF-HFP) before this electric field. It also indicates that the loss is lower when there is low residual polarization at high field strength. When the electric fields rise over 200 kV/cm, field-induced phase transformation occurs, the residual polarization values dramatically grow, and it tends towards stability over 500 kV/cm. It can also be seen from Figure 6 that at 200~500 kV/mm, the residual polarization values of 3-0-3 are slightly lower than in other samples. This phenomenon is relative to the reduction in ferroelectric loss, which corresponds to the polymer’s decreased loss when the insulating nanosheets (2D-Al_2_O_3_) are well mixed into polymers.

To sum up, the ×0× nanocomposites’ energy storage density is closely related to their electric field breakdown strength. Normally, the electric field breakdown strength depends on many parameters, and the electrical tree is one of the main causes [37,38]. Electrical trees consist of many gas channels; these gas channels are caused by many factors, for instance, structure defects (voids and pores), partial discharge activity, protrusions from the electrodes, conducting particles, and so on [39]. These defects can be suppressed or eliminated by improving the preparation process of nanocomposites. Then, the insulation property is improved to enhance the energy storage density of ×0× nanocomposites.

Here, the ×0× structure is designed to restrain the nanocomposites’ breakdown effect, which is evidenced by measuring the leakage current at high field strengths (Figure 7). When the field strength is above 200 kV/mm, field-induced phase transformation occurs, and the leakage current of pure P(VDF-HFP) is gradually higher than 2 × 10^−6^ A/cm^2^. When a small volume ratio of 2D-Al_2_O_3_ nanosheets is added into pure P(VDF-HFP), a noteworthy feature of ×0× nanocomposites is that the leakage current values tend to decrease, which indicates that the good insulation feature of ×0× nanocomposites come from the 2D-Al_2_O_3_ nanosheets and the sandwich structure synergistic contributions. In previous research studies, large contents of dielectric nanoparticles were mixed into polymer matrices with the aim of increasing the polarization or breakdown strength [40]. Nevertheless, nanocomposites’ energy storage properties decreased by the large contents of nanoparticles exceeding the permeation threshold [41]. The interesting thing in this work is that when changing the spatial distribution of nanosheets to the interfacial region, the content of nanosheets will be much lower than the percolation threshold, while the nanocomposites keep good properties.

In addition to the insulating test, we also conducted a simulation of the ×0× nanocomposites’ spatial distribution of the electric field. As shown in Figure 8, it provides a deeper understanding of the origin of suppressed leakage in nanocomposites. Nanocomposites’ microstructure significantly impacts the breakdown strength and breakdown path [35]. The charge carriers will have a longer scattering path when they encounter nanosheets; the enhanced electrical performance of nanocomposites is related to those scattered charges and homocharges (which are generated at the electrodes) [42,43]. The function of the homocharge is to block the further charge injection; therefore, the voltage is increased [43,44]. The simulation results revealed that 2D-Al_2_O_3_ nanosheets block the flow of electric current between the electrodes. Because of their large isolating interfacial areas that create a steric hindrance effect against the growth of electrical trees, the ×0× nanocomposite layers near the electrode act as buffer layers to bear the electric field. This leads to not only high breakdown strengths but also the suppression of leakage current density (i.e., conduction loss) at high fields. These two improvements synergistically contribute to the high energy performance of these nanocomposites.

## 4. Conclusions

In summary, this work demonstrates the feasibility of designing ultra-thin insulating nanosheets and artificial sandwich structures for high-performance dielectric materials. P(VDF-HFP) as a polar polymer with a small addition (1–7 vol%) of 2D-Al_2_O_3_ nanosheets as the buffer layers could dramatically increase the electric field breakdown strength by 24.8%, which is about 740 kV/mm, while the energy density is improved by 30.6%, which is about 23.5 J/cm^3^ in P(VDF-HFP)/2D-Al_2_O_3_ nanocomposite (with 3 vol.% nanosheets). This work not only proposes new choices for modulating energy storage properties in polar composites but also demonstrates the potential of polar polymer dielectrics to realize high electric breakdown strength comparable to those of non-polar polymers, given that a wide variety of nanosheets are available for nanostructure modulating. The further directions of the research are to explore the mechanism effect of different nanosheets and multilayer structures on P(VDF-HFP) nanocomposite.

## Figures and Tables

**Figure 1 nanomaterials-13-02836-f001:**
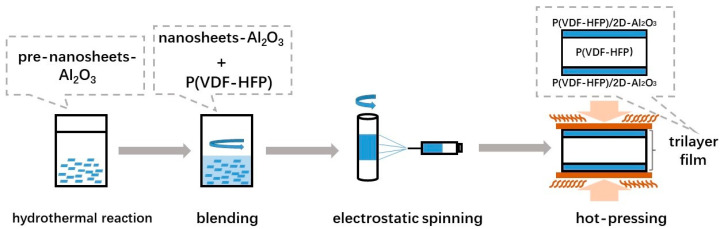
Preparation process diagram of ultra-thin Al_2_O_3_ nanosheets (2D-Al_2_O_3_) and P(VDF-HFP)/2D-Al_2_O_3_ nanocomposites.

**Figure 2 nanomaterials-13-02836-f002:**
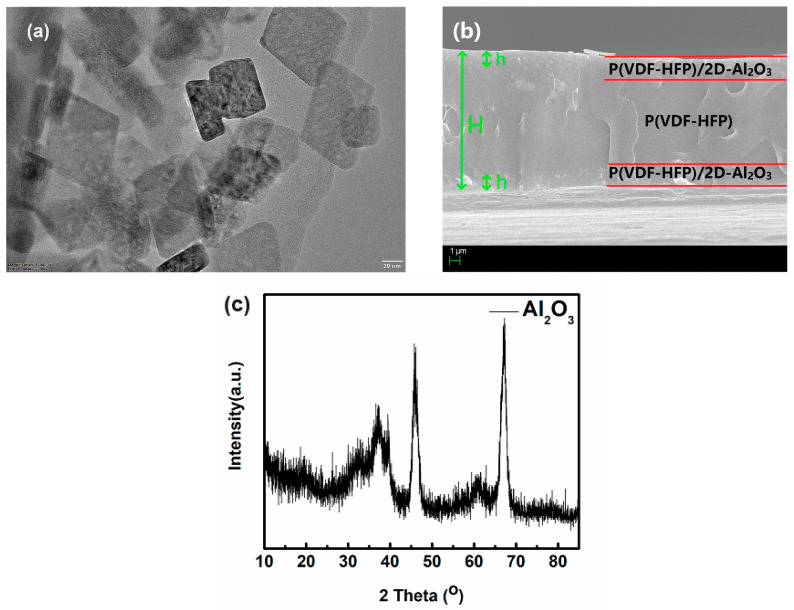
(**a**) HRTEM images of 2D-Al_2_O_3_ nanosheets; (**b**) cross-sectional SEM images of P(VDF-HFP)/2D-Al_2_O_3_ nanocomposites; and (**c**) XRD images of 2D-Al_2_O_3_ nanosheets.

**Figure 3 nanomaterials-13-02836-f003:**
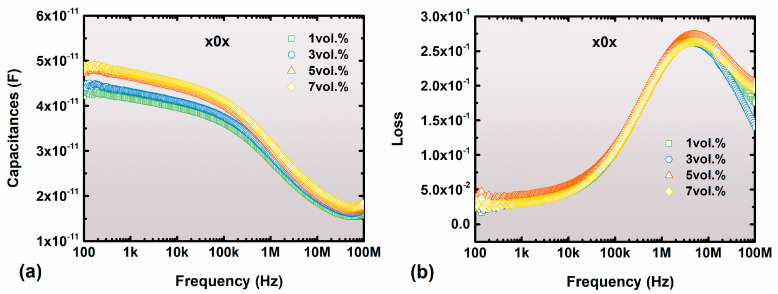
(**a**) The capacitances and (**b**) dielectric loss (tan δ) as a function of frequency at 25 °C for the P(VDF-HFP)/2D-Al_2_O_3_ nanocomposites.

**Figure 4 nanomaterials-13-02836-f004:**
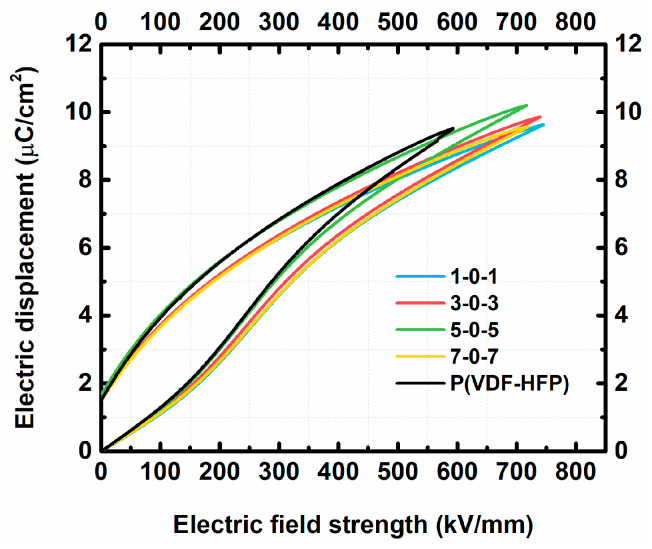
Electric displacement-electric field (D-E) loops of the ×0× nanocomposites and P(VDF-HFP) matrix.

**Figure 5 nanomaterials-13-02836-f005:**
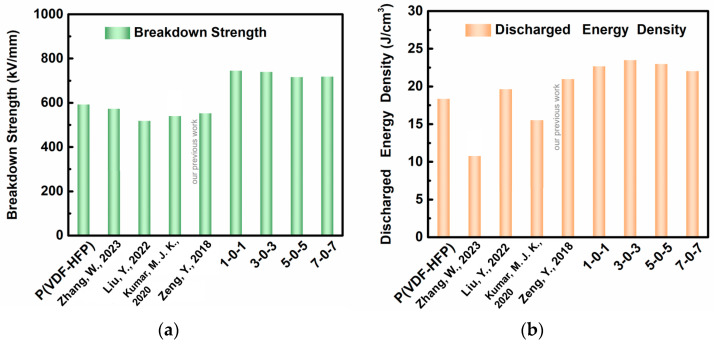
Comparisons between (**a**) breakdown strength and (**b**) discharged energy density of ×0× nanocomposites, P(VDF-HFP) matrix, and other nanocomposites [23,24,26,30].

**Figure 6 nanomaterials-13-02836-f006:**
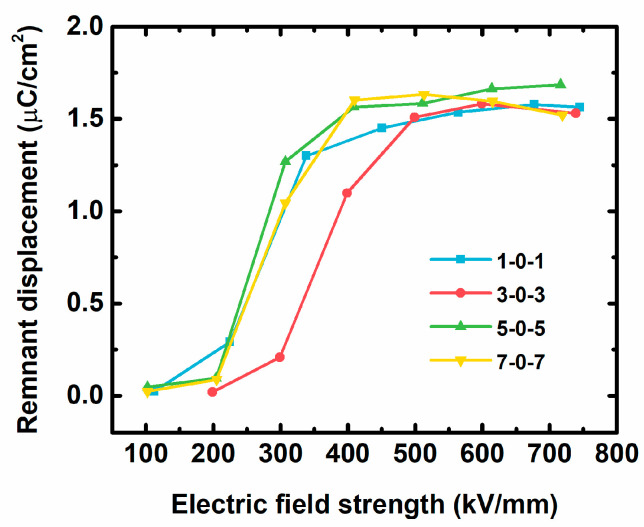
The remnant displacement values of ×0× nanocomposites at different electric field strengths.

**Figure 7 nanomaterials-13-02836-f007:**
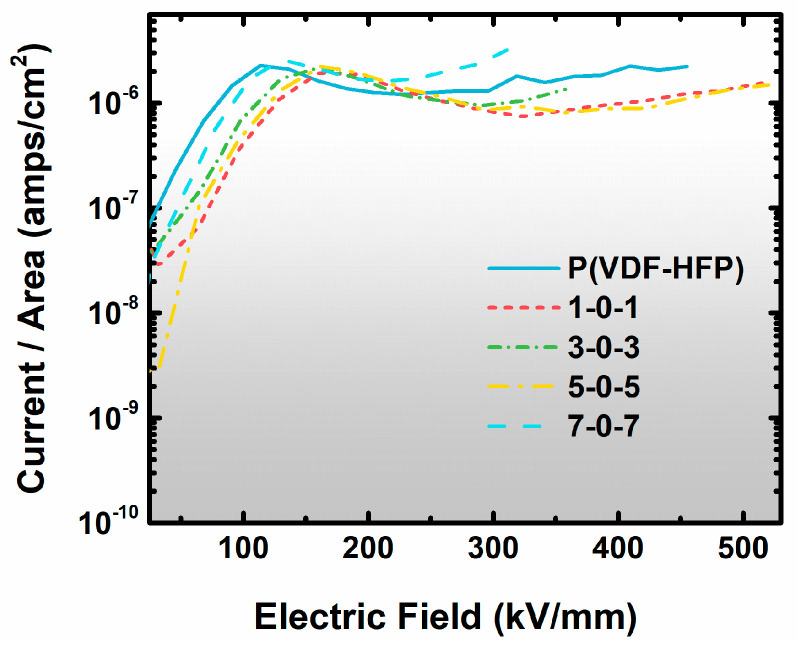
The leakage current of ×0× nanocomposites and P(VDF-HFP) matrix at high field strengths.

**Figure 8 nanomaterials-13-02836-f008:**
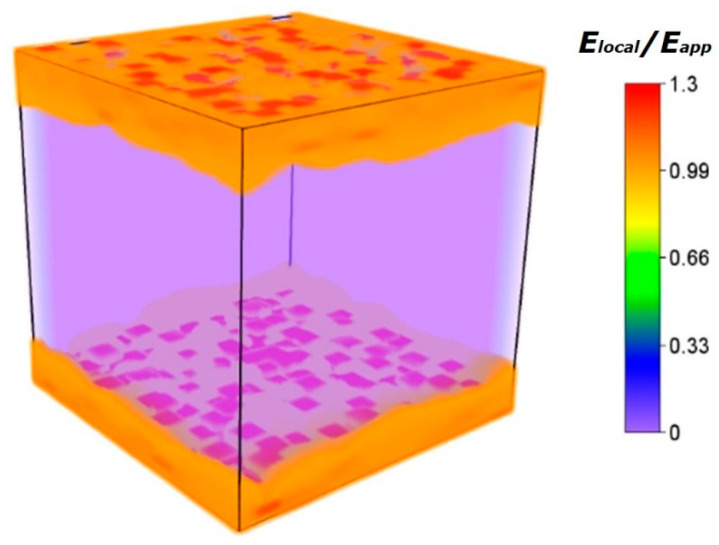
Phase-field simulation of local electric field distribution of P(VDF-HFP)/2D-Al_2_O_3_ nanocomposite with sandwich structure.

## Data Availability

The data presented in this study are available on request from the corresponding author.

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
