# Peer review of "Improved Breakdown Strength and Restrained Leakage Current of Sandwich Structure Ferroelectric Polymers Utilizing Ultra-Thin Al_2_O_3_ Nanosheets"

_nanomaterials, 2023, doi:10.3390/nano13212836_

Round 1
Reviewer 1 Report
The authors describe a novel nanocomposite approach to improve the electrical properties of ferroelectric polymers. This is potentially useful in many flexible electronic applications. My main concern is reproducibility and experimental error. How reproducible is the fabrication process? What is the variability of the resulting nanostructures? All the graphs don't contain error bars. In some cases these may be minor but then comments about the numerical size of the experimental errors should be included.
Some minor editing by an editorial assistant before publication may help legibility.
Author Response
Dear reviewer:
I am very grateful to your comments for the manuscript. According with your advice, we amended the relevant part in manuscript. Some of your questions were answered below.
Q1: How reproducible is the fabrication process? What is the variability of the resulting nanostructures?
Response:To be more clearly and in accordance with the reviewer concerns, we have added a more detailed interpretation regarding repeatability of manufacturing processes and stability of nanostructures. All the preparation process is repeated more than three times, the structure of the calcined nanosheets is very stable because it forms a gamma phase of Al2O3. More details were added on page 2 line 83-86, page 3 line 100-101.
Q2: All the graphs don't contain error bars. In some cases, these may be minor but then comments about the numerical size of the experimental errors should be included.
Response:We are grateful for the suggestion. Because the curves we test are usually tested with a continuous frequency or electric field, it is not possible to add error bars to the curves. Each composite films include over 30 tested points to achieve repeatability of properties. The defects introduced during sputtering copper electrodes or preparation of composites will cause the low properties of composites,however,this probability is approximately less than 5 percent. Changes highlighted in red on page 3 line 121-124.
Reviewer 2 Report
The paper is focused on preparation of the composites (sandwiche structures) based on ferroelectric polymers and Al2O3 nanosheets. The paper topic is timely because improvement of breakdown strength and energy density is the important issue for commercial systems. Overall the text is correctly organized however several points need to be improved before publishing:
1. The language should be improved (there are several misspelling errors)
2. The half of page 3 have been copied from ref. [31]
3. The figures should be improved (e.g. 2a – same results of the TEM investigation are covered by the XRD)
4. The fig 3a and 3b – the differences are not clearly visible.
5. Some part from the “results and discussion” should be moved to the “introduction”
line 32
line 67
line 41
line86
line 88
Author Response
Dear reviewer:
I am very grateful to your comments for the manuscript. According with your advice, we amended the relevant part in manuscript. Some of your questions were answered below.
Q1. The language should be improved (there are several misspelling errors). Comments on the Quality of English Language,line 32,line 67,line 41,line86,line 88.
Response:We are grateful for the suggestion, changes highlighted in red have been made accordingly in the revised manuscript.
Q2. The half of page 3 have been copied from ref. [31]
Response:There is a misunderstanding about the site of cited reference. We cite this reference to explain how to use the theory of phase field in our composites. We change the site of cited reference to avoid misunderstanding (page 3, line 127, ref 35).
Q3. The figures should be improved (e.g. 2a – same results of the TEM investigation are covered by the XRD)
Response:This has been clarified in the revised version of the manuscript, XRD image has listed separately as fig 2c in new Figure 2. (page 4-5, line141-142, line 152-157).
Q4. The fig 3a and 3b – the differences are not clearly visible.
Response:The test results indicate the magnitude order of different composites' capacitances or dielectric loss are in a same magnitude order, thus there are less discrepancy between different composites.
Q5. Some part from the “results and discussion” should be moved to the “introduction”
Response:Changes highlighted in red have been made accordingly in the revised manuscript (page 2, line 53-64).
Reviewer 3 Report
The paper presents investigation of the P(VDF-HFP) with ultra-thin Al2O3 nanosheets composite films. Materials was fabricated through an electrospinning process followed by hot-pressing in sandwich structured. The research was conducted in the context of the use of flexible capacity applications demands to large energy storage density and high breakdown electric field strength of flexible films. The results show that the insulating ultra-thin Al2O3 nanosheets and the sandwich structure can enhance the composites' breakdown strength and energy density remarkedly to P(VDF-HFP) polymer matrix. In my opinion this paper can be interesting to readers of Nanomaterials journal English of the paper is rather good and meet the requirement of the journal – in my opinion the language of the paper should be a little improved. The manuscript can be accepted for publication after MINOR corrections.
For improving the quality of the paper, the authors should address the following comments:
- Introduction chapter should be correct. Authors should include new information about topic of a paper. Amount of references is also sufficient but some papers cited in the references (30 from all 42) are older then 5 years. It would be desirable to expand this list somewhat by adding the work of other authors in the field of research over the past five years. That can help to emphasize the relevance and significance of this study.
- Moreover Authors should include several modern papers of global research in this field – more information based on worldwide (global) study – not mostly from Asia (minimum 32 references). Authors should include several modern papers (also from Europe and America).
- Please refer to the scientific achievements of research teams from around the world related to the topic of the paper. Please see on the papers on nanocomposites based on Al2O3 dielectrics and their properties of professors A.K. Fedotov and/or P. Zukowski.
- The article presents several previous research works in the field but lacks a detailed summary evaluation and identification of existing issues. To better highlight the novelty and argument of this paper, it is recommended that the Authors establish a stronger connection between the current work and the problems identified in the previous research (applies to research conducted at their facility and internationally).
- How do the Authors see the prospects for practical application of the results obtained during the performed research?
- What further directions of the initiated research and possibilities of its practical application?
- Please prepare a literature list according to the guidelines of the Nanomaterials journal (MDPI Publisher).
The presented paper suits the requirements of the Nanomaterials journal and it may be publish after MINOR corrections.
In my opinion this paper can be interesting to readers of Nanomaterials journal English of the paper is rather good and meet the requirement of the journal – in my opinion the language of the paper should be a little improved.
Author Response
Dear reviewer:
I am very grateful to your comments for the manuscript. According with your advice, we amended the relevant part in manuscript. Some of your questions were answered below.
Q1. Introduction chapter should be correct. Authors should include new information about topic of a paper. Authors should include several modern papers (also from Europe and America). Please refer to the scientific achievements of research teams from around the world related to the topic of the paper. Please see on the papers on nanocomposites based on Al2O3 dielectrics and their properties of professors A.K. Fedotov and/or P. Zukowski. Please prepare a literature list according to the guidelines of the Nanomaterials journal.
Response:We are grateful for the suggestion. Some new references were added on page 1-2, line 30, line 48-51, line 62-64. (ref 2, ref 3, ref 15, ref 16, ref 18, ref 21, ref 22, ref 25).
Q2. How do the Authors see the prospects for practical application of the results obtained during the performed research? What further directions of the initiated research and possibilities of its practical application?
Response:To be more clearly and in accordance with the reviewer suggestions, changes highlighted in red have been made accordingly in the revised manuscript (page 8-9, line 268-273).
Reviewer 4 Report
Following comments to improve the manuscript for further processing,
1. In Fig.2 SEM image please add thickness measurements and add the same details in discussion.
2. In addition individual layer thickness must be included and surface morphological SEM images which work as a good reference to compare with the other methods.
3. In line 244 what is meant by x0x? Check and explain.
4. Data figures from Fig.4 to Fig. 7 should be replotted and revised for better quality images. Legends are repeated and of poor quality.
Author Response
Dear reviewer:
I am very grateful to your comments for the manuscript. According with your advice, we amended the relevant part in manuscript. Some of your questions were answered below.
Q: In Fig.2 SEM image please add thickness measurements and add the same details in discussion. In addition, individual layer thickness must be included and surface morphological SEM images which work as a good reference to compare with the other methods.
Response:The indicating arrow of thickness were add in fig2, the same details add in line 146-147. Cross-sectional SEM image is the normal technology in materials research area to show the structure of multilayer film.
Q: In line 244 what is meant by x0x? Check and explain.
Response:The word “x0x” first occur in section 2.1 “Preparation of Samples, line 90-94 in original manuscript, line 102-106 in revised manuscript.
Q: Data figures from Fig.4 to Fig. 7 should be replotted and revised for better quality images. Legends are repeated and of poor quality.
Response:Figures were prepared with the demand of “nanomaterials” (a resolution of 300 dpi or higher). The description conforms to the scientific terminology specification.
Round 2
Reviewer 4 Report
The authors responded to all the comments with relevant discussion, highlighted all the revised manuscript changes, and improved accordingly.